# Selective optogenetic activation of Na$_V$1.7–expressing afferents in Na$_V$1.7-ChR2 mice induces nocifensive behavior without affecting responses to mechanical and thermal stimuli

**Toyoaki Maruta**[1]\*, **Kotaro Hidaka**[1], **Satoshi Kouroki**[1], **Tomohiro Koshida**[1], **Mio Kurogi**[1], **Yohko Kage**[2], **Seiya Mizuno**[3], **Tetsuro Shirasaka**[1], **Toshihiko Yanagita**[4], **Satoru Takahashi**[3], **Ryu Takeya**[2], **Isao Tsuneyoshi**[1]

**1** Department of Anesthesiology, Faculty of Medicine, University of Miyazaki, Miyazaki, Miyazaki, Japan,
**2** Department of Pharmacology, Faculty of Medicine, University of Miyazaki, Miyazaki, Miyazaki, Japan,
**3** Laboratory Animal Resource Center in Transborder Medical Research Center, Faculty of Medicine, University of Tsukuba, Tsukuba, Ibaraki, Japan, **4** Department of Clinical Pharmacology, School of Nursing, Faculty of Medicine, University of Miyazaki, Miyazaki, Miyazaki, Japan

\* mmctm2@yahoo.co.jp

## Abstract

In small and large spinal dorsal root ganglion neurons, subtypes of voltage-gated sodium channels, such as Na$_V$1.7, Na$_V$1.8, and Na$_V$1.9 are expressed with characteristically localized and may play different roles in pain transmission and intractable pain development. Selective stimulation of each specific subtype *in vivo* may elucidate its role of each subtype in pain. So far, this has been difficult with current technology. However, Optogenetics, a recently developed technique, has enabled selective activation or inhibition of specific neural circulation *in vivo*. Moreover, optogenetics had even been used to selectively excite Na$_V$1.8-expressing dorsal root ganglion neurons to induce nocifensive behavior. In recent years, genetic modification technologies such as CRISPR/Cas9 have advanced, and various knock-in mice can be easily generated using such technology. We aimed to investigate the effects of selective optogenetic activation of Na$_V$1.7-expressing afferents on mouse behavior. We used CRISPR/Cas9-mediated homologous recombination to generate bicistronic Na$_V$1.7–iCre knock-in mice, which express iCre recombinase under the endogenous Na$_V$1.7 gene promoter without disrupting Na$_V$1.7. The Cre-driver mice were crossed with channelrhodopsin-2 (ChR2) Cre-reporter Ai32 mice to obtain Na$_V$1.7$^{iCre/+}$;Ai32/+, Na$_V$1.7$^{iCre/iCre}$;Ai32/+, Na$_V$1.7$^{iCre/+}$;Ai32/Ai32, and Na$_V$1.7$^{iCre/iCre}$;Ai32/Ai32 mice. Compared with wild–type mice behavior, no differences were observed in the behaviors associated with mechanical and thermal stimuli exhibited by mice of the aforementioned genotypes, indicating that the endogenous Na$_V$1.7 gene was not affected by the targeted insertion of iCre. Blue light irradiation to the hind paw induced paw withdrawal by mice of all genotypes in a light power-dependent manner. The threshold and incidence of paw withdrawal and aversive behavior in a blue-lit room were dependent on ChR2 expression level; the

**Data Availability Statement:** All relevant data are within the paper and its Supporting Information files.

**Funding:** Toyoaki Maruta was supported by Japan Society for the Promotion of Science (https://www.jsps.go.jp/english/index.html): KAKENHI Grant Numbers JP18K08859, 21K08925, and 16H06276 (Advanced Animal Model Support, http://model.umin.jp/english/index.html). The funders had no role in study design, data collection and analysis, decision to publish, or preparation of the manuscript.

**Competing interests:** The authors have declared that no competing interests exist.

strongest response was observed in $Na_V1.7^{iCre/iCre}$;Ai32/Ai32 mice. Thus, we developed a non-invasive pain model in which peripheral nociceptors were optically activated in free-moving transgenic $Na_V1.7$–ChR2 mice.

# Introduction

The sensation of pain results from the activation of a subset of sensory neurons known as nociceptors. Activation of unmyelinated (C-fiber) and myelinated (Aδ-fiber) nociceptive afferent fibers indicates potential tissue damage, which is reflected in the high thresholds of nociceptors for mechanical, thermal, and chemical stimuli, which induce neurotransmissions via ion channels, neurotransmitters, and intracellular signaling [1, 2]. These conditions change considerably in neuropathic pain states. Understanding the changes that occur in neuropathic pain is vital for identifying new therapeutic targets and developing novel analgesics [2]. However, since mechanical, thermal, and chemical stimuli activate both targeted and off-target neurons, effectively controlling the activation of specific types of neurons is a major challenge.

Optogenetics is a recently developed and popular tool used in several areas of neuroscience research [3]. This technique utilizes light-sensitive ion channels (opsins) to modulate the activity of specific neuron subsets. Several types of opsins are currently used, with new ones being continuously developed and optimized. This technique, which enables selective activation or inhibition of neural circulation *in vivo*, can be used to provide a better understanding of complex pain pathways [4–7].

Channelrhodopsins are a subfamily of retinylidene proteins (rhodopsins) that function as light–gated ion channels. They are nonspecific cation channels that conduct $H^+$, $Na^+$, $K^+$, and $Ca^{2+}$ ions. Channelrhodopsin-1 (ChR1) and Channelrhodopsin-2 (ChR2) from the model organism *Chlamydomonas reinhardtii* were the first discovered channelrhodopsins. Recently, various ChR2 transgenic animal models have been generated, which play important roles in revealing the mechanisms of neural activity and mapping neural circuits [8].

Different subtypes of voltage-gated sodium channels, including $Na_V1.7$, $Na_V1.8$, and $Na_V1.9$, exhibit characteristic localization in neurons of the small and large spinal dorsal root ganglion (DRG) and may play different roles in pain transmission and neuropathic pain development [9, 10]. If a specific subtype of voltage-gated sodium channels could be selectively stimulated in a living body, it might be possible to evaluate its specific role in pain. However, this has been difficult with current technology. Optogenetics enables selective activation or inhibition of neural circulation *in vivo*. The first voltage-dependent $Na^+$ channel to be targeted in pain studies using ChR2 transgenic animals was $Na_V1.8$ [11, 12]. However, transgenic mice wherein other $Na^+$ channels are targeted have not been produced. Currently, genetic modification technologies such as CRISPR/Cas9 have advanced, and various knock-in mice can be easily generated. The present study aimed to examine mouse nocifensive behavior as a result of selective optogenetic activation of $Na_V1.7$–expressing afferents. This study helps elucidate the role of $Na_V1.7$ in pain neurotransmission and might provide valuable insights into the development of neuropathic pain and the design of effective therapeutics in the future.

# Results

## Development of original $Na_V1.7$–iCre ($Na_V1.7^{iCre/+}$) knock-in mice

The founder generation (F0) of bicistronic $Na_V1.7$–iCre ($Na_V1.7^{iCre/+}$) knock-in mice was created using the CRISPR/Cas9 system (Fig 1A), with s assistance from the platform of Advanced

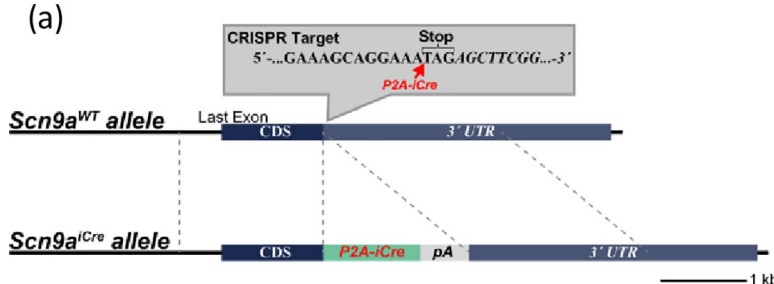

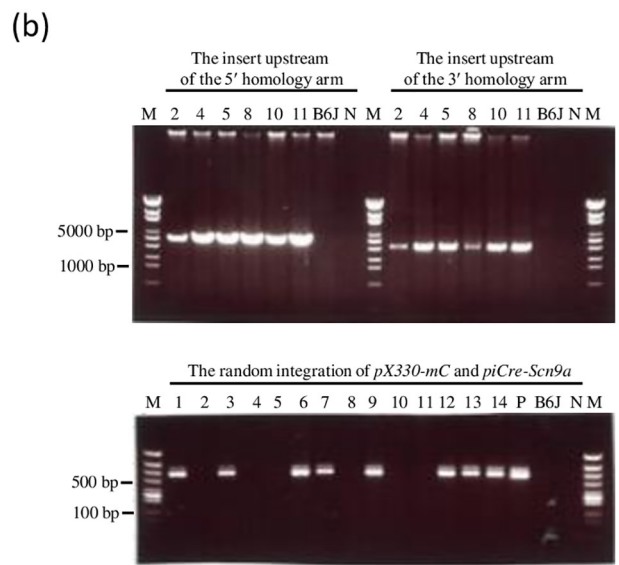

**Fig 1.** Schematic diagram of the *Scn9a^iCre* knock-in allele (a) and PCR for genotyping founder mice (b). (a)The last exon of *Scn9a* is illustrated. The P2A-iCre sequence was inserted immediately after the stop codon of *Scn9a*. (b) The detection of the insert upstream of the 5′- and 3′-homology arm and the random integration of *pX330-mC* and *piCre-Scn9a*. Six founder mice were determined to carry the designed knock-in mutation. M, marker; P, positive control; N, negative control.

Animal Model Support (AdAMS). The sequence 5′-gaaagcaggaaatagagctt-3′ containing the termination codon of *Scn9a* was selected as the target for the single guide RNA (sgRNA) and inserted into the *pX330-mC* plasmid, which carried both sgRNA and *Cas9-mC* expression units [13]. A P2A-iCre-rabbit globin polyadenylation sequence was present between the 5′- and 3′- homology arms of the donor DNA, *piCre-Scn9a*. P2A is a 2A self-cleaving peptide that helps generate polyproteins by preventing the ribosome from creating a peptide bond (ribosome skipping). The genomic region from 1,656 to 1 bp upstream of the termination codon was isolated in the 5′-homology arm, whereas the genomic region from the termination codon to 2,054 bp downstream of the termination codon was isolated in the 3′-homology arm of *Scn9a*. These DNA vectors were isolated using the FastGene Plasmid Mini Kit (Nippon Genetics, Tokyo, Japan) and filtered through a Millex-GV® filter (0.22 μm; Merck Millipore, Darmstadt, Germany).

Pregnant mare serum gonadotropin (5 IU) and human chorionic gonadotropin (5 IU) were intraperitoneally injected into female C57BL/6J mice at 48-h intervals. Subsequently, they mated with male C57BL/6J mice. Afterwards., we obtained the zygotes from the oviducts of the mated females and microinjected a mixture of *pX330-mC* (circular, 5 ng/μL, each) and

*piCre-Scn9a* (circular, 10 ng/μL) into the zygotes. Subsequently, the surviving zygotes were transferred into the oviducts of pseudopregnant ICR strain females mice of the ICR strain, and newborns were obtained. To confirm the presence of the knock-in mutation, we purified the genomic DNA isolated from a tail tissue sample was purified using a PI-200 DNA automatic isolation system (Kurabo Industries Ltd., Osaka, Japan) according to the manufacturer's instructions. Genomic PCR was performed using a KOD Fx kit (TOYOBO, Osaka, Japan). The insert upstream of the 5′ homology arm was detected using the following primers: *Scn9a* screening 5Fw (5′- `TGATGGCCATAAAAATCAAAGGATGGTA`-3′) and iCre-5-Rv68 (5′- `AGATCCATCTCTCCACCAGCTTGGTAAC` -3′). The insert upstream of the 3′-homology arm was detected using the following primers: iCre-3-Fw68 (5′- `GAGGATGTGAGGGACTACCTC` `CTGTACC` -3′) and *Scn9a* screening 3Rv (5′- `GGATGTTTTGTGTGGCTCACCATTAAGT`-3′). Six founder mice were determined to carry the designed knock-in mutation. Additionally, we performed PCR to detect random integration of *pX330-mC* and *piCre-Scn9a* with a primer that detected the ampicillin resistance gene (Amp detection-F: 5′-`ttgccgggaagctagag``taa`-3′, and Amp detection-R: 5′-`tttgccttcctgttttttgct`-3′). No founder mice was determined to carry the random integration allele. Fig 1B shows the results of genomic PCR.

## Development of Na$_V$1.7-ChR2 mice genotypes

Fig 2A shows the mating patterns of mice adopted the following four genotypes of Na$_V$1.7-ChR2 mice: Na$_V$1.7$^{iCre/+}$;Ai32/+, Na$_V$1.7$^{iCre/iCre}$;Ai32/+, Na$_V$1.7$^{iCre/+}$;Ai32/Ai32, and Na$_V$1.7$^{iCre/iCre}$;Ai32/Ai32. We created Na$_V$1.7$^{iCre/+}$;Ai32/+ mice by crossing homozygous

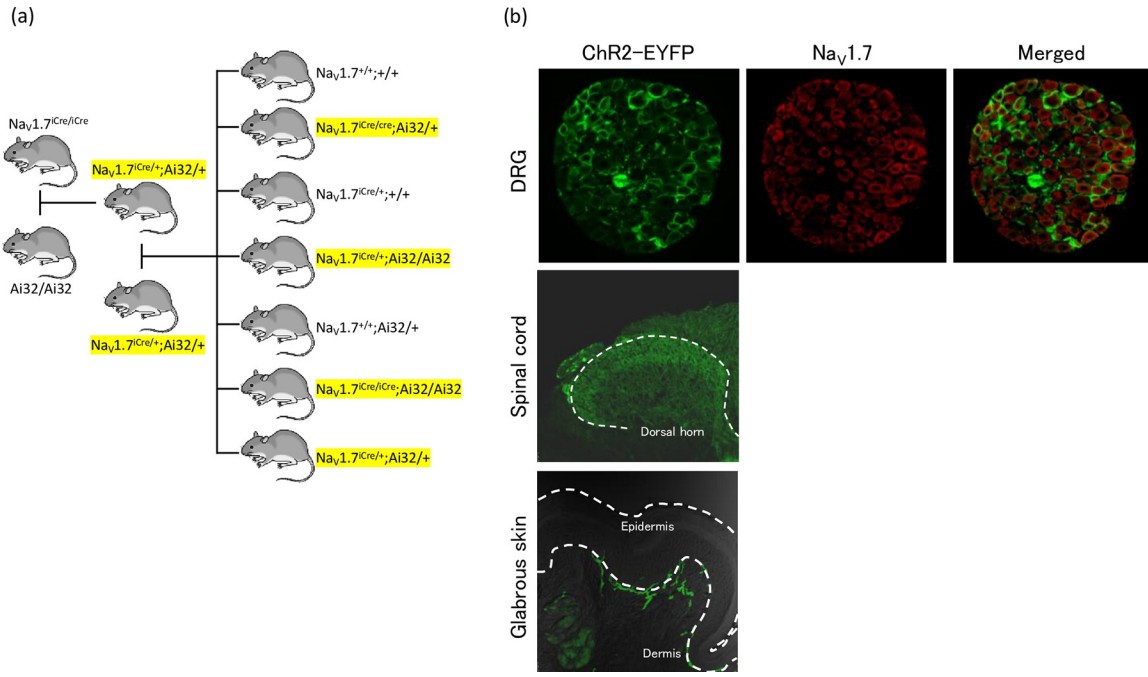

**Fig 2.** Establishment of different lines of transgenic mice (a) and distribution of ChR2-EYFP channels in DRG (b). (a) Na$_V$1.7$^{iCre/+}$; Ai32/+ mice were created by crossing homozygous Na$_V$1.7–iCre mice with heterozygous Ai32 mice. Subsequently, Na$_V$1.7$^{iCre/+}$;Ai32/+, Na$_V$1.7$^{iCre/iCre}$;Ai32/+, Na$_V$1.7$^{iCre/+}$;Ai32/Ai32, and Na$_V$1.7$^{iCre/+}$;Ai32/Ai32 mice were created by crossing Na$_V$1.7$^{iCre/+}$;Ai32/+ mice with each other. The four genotypes of mice used in the study are highlighted in yellow. (b) A typical immunohistochemical image showing of ChR2-EYFP in the DRG, the dorsal horn of spinal cord, and glabrous skin of Na$_V$1.7$^{iCre/+}$;Ai32/+ mouse was shown. Green and red fluorescence indicates ChR2-EYFP and Na$_V$1.7, respectively. ChR2–EYFP were expressed on Na$_V$1.7-expressing DRG neurons. Green fluorescence (ChR2–EYFP expression) can be observed in the dorsal horn. ChR2–EYFP is localized in free nerve endings in the lower and upper dermis of glabrous skin.

Na$_V$1.7–iCre (Na$_V$1.7$^{iCre/iCre}$) mice with homozygous Ai32 mice carrying the *ChR2(H134R)-EYFP* gene in their *Gt(ROSA)26Sor* locus [14]. The gene was separated from its CAG promoter using a loxP-flanked transcriptional STOP cassette, allowing its expression in a Cre-dependent manner. Thereafter, Na$_V$1.7$^{iCre/+}$;Ai32/+, Na$_V$1.7$^{iCre/iCre}$;Ai32/+, Na$_V$1.7$^{iCre/+}$;Ai32/Ai32, and Na$_V$1.7$^{iCre/iCre}$;Ai32/Ai32 mice were created by crossing Na$_V$1.7$^{iCre/+}$;Ai32/+ mice with each other.

Fig 2B shows a typical immunohistochemical image showing Na$_V$1.7 and ChR2-EYFP in the DRG of Na$_V$1.7$^{iCre/+}$;Ai32/+ mouse. EYFP fluorescence was observed in Na$_V$1.7-expressing DRG neurons. EYFP fluorescence was also observed in the dorsal horn of the spinal cord and was faded in the deep and superficial layers of the glabrous skin bordering the dermal–epidermal junction.

## Normal nocifensive behavior associated with mechanical and thermal stimuli was exhibited by the four genotypes

We examined the behavior associated with mechanical and thermal stimuli exhibited by four genotypes of Na$_V$1.7-ChR2 mice. As shown in Fig 3A and 3B, compared with the behavior of wild-type (WT) mice, no differences were observed in the behavior associated with mechanical and thermal stimuli exhibited by the mice of the four genotypes.

## Differences among the four genotypes in behavior associated with optogenetic stimulation

We performed two types of behavioral tests using optogenetics. Fig 4 shows a schematic representation of the blue LED light irradiation hind paw withdrawal test. The mice of the four genotypes showed a light power-dependent increase in withdrawal percentage and sensitivity to blue light in the following order: Na$_V$1.7$^{iCre/+}$;Ai32/+ < Na$_V$1.7$^{iCre/iCre}$;Ai32/+ < Na$_V$1.7$^{iCre/+}$;Ai32/Ai32 < Na$_V$1.7$^{iCre/iCre}$;Ai32/Ai32. To rule out a ChR2-independent effect of strong illumination on the animal behavior, we performed a yellow LED light irradiation hind paw withdrawal test on Na$_V$1.7$^{iCre/+}$;Ai32/+ mice to rule out a ChR2-independent effect of strong illumination on the animal behaviour. Irradiation with yellow light irradiation (5 mW) did not induce hind paw withdrawal behavior.

Fig 5 shows a schematic representation of the optogenetic place aversion (OPA) test. The time spent in the room with the blue LED floor was shorter for mice of all four genotypes than that for WT mice. The duration of stay in the room by mice decreased as follows: Na$_V$1.7$^{iCre/+}$;Ai32/+ = Na$_V$1.7$^{iCre/iCre}$;Ai32/+ > Na$_V$1.7$^{iCre/+}$;Ai32/Ai32 = Na$_V$1.7$^{iCre/iCre}$;Ai32/Ai32.

## Expression of ChR2 in DRG of mice of the four genotypes

Fig 6 shows ChR2 expression in the DRG of mice of the four genotypes as measured using reverse transcription-polymerase chain reaction (RT–PCR). The expression of ChR2 in mice of the four genotypes increased in the following order: Na$_V$1.7$^{iCre/+}$;Ai32/+ ≤ Na$_V$1.7$^{iCre/iCre}$;Ai32/+ < Na$_V$1.7$^{iCre/+}$;Ai32/Ai32 = Na$_V$1.7$^{iCre/iCre}$;Ai32/Ai32. S2 Fig shows the typical immunohistochemical image showing ChR2-EYFP expression in the DRG of mice of the four genotypes. A higher number of EYFP-positive DRG neurons were observed in Na$_V$1.7$^{iCre/+}$;Ai32/Ai32 and Na$_V$1.7$^{iCre/iCre}$;Ai32/Ai32 mice than that in Na$_V$1.7$^{iCre/+}$;Ai32/+ mice.

## Discussion

We used CRISPR/Cas9-mediated homologous recombination to generate bicistronic Na$_V$1.7–iCre knock-in mice expressing iCre recombinase under the control of the endogenous Na$_V$1.7

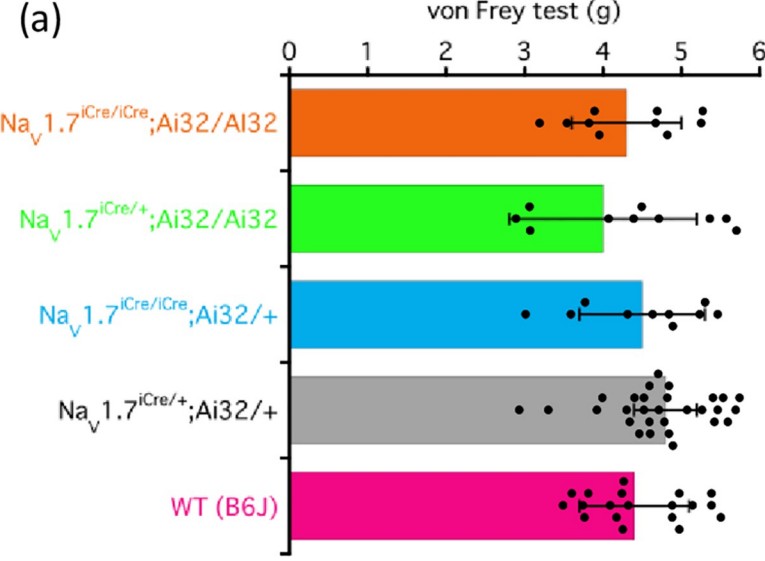

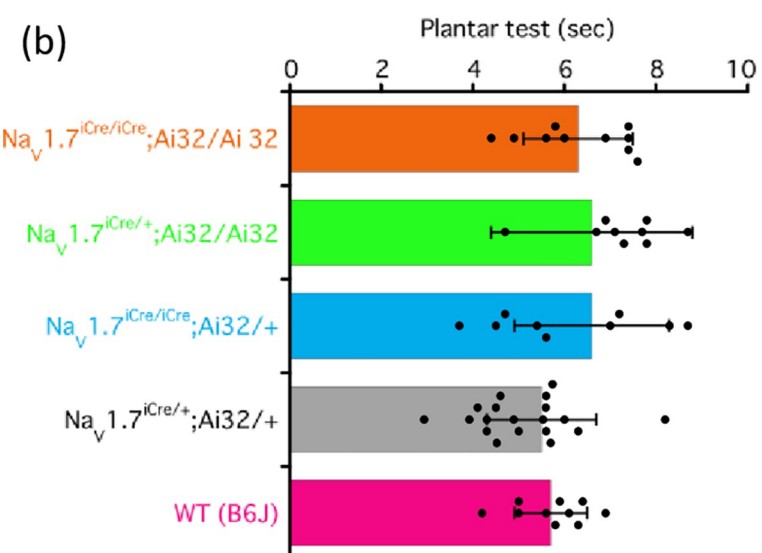

**Fig 3.** Paw withdrawal test (von Frey test) (a) and plantar test (b). (a) The von Frey test was performed with the wild type (WT) and mice of the four genotypes. The hind paw withdrawal data were analyzed using one-way ANOVA. All results are calculated as mean ± SD of 10 or more animals. Individual results for each strain are as follows: WT (B6J), 4.4 ± 0.7 g; $Na_V1.7^{iCre/+}$;Ai32/+, 4.8 ± 0.4 g; $Na_V1.7^{iCre/iCre}$;Ai32/+, 4.5 ± 0.8 g; $Na_V1.7^{iCre/+}$;Ai32/Ai32, 4.0 ± 1.2 g; and $Na_V1.7^{iCre/iCre}$;Ai32/Ai32, 4.3 ± 0.7 g. (b) The plantar test was performed with the WT and mice of the four genotypes. The data were analyzed using one-way ANOVA. All results are calculated as mean ± SD of 10 or more animals. Individual results for each strain were as follows: WT (B6J), 5.7 ± 0.8 s; $Na_V1.7^{iCre/+}$;Ai32/+, 5.5 ± 1.2 s; $Na_V1.7^{iCre/iCre}$; Ai32/+, 6.6 ± 1.7 s; $Na_V1.7^{iCre/+}$;Ai32/Ai32, 6.6 ± 2.2 s; and $Na_V1.7^{iCre/iCre}$;Ai32/Ai32, 6.3 ± 1.2 s.

gene promoter without disrupting endogenous $Na_V1.7$. Furthermore, using the Cre-loxP system, we crossed homozygous $Na_V1.7$–iCre ($Na_V1.7^{iCre/iCre}$) mice with homozygous Ai32 mice and generated a transgenic $Na_V1.7$–ChR2 mouse line ($Na_V1.7^{iCre/+}$;Ai32/+) in which ChR2 was selectively targeted in $Na_V1.7$-expressing sensory neurons. Thus, the $Na_V1.7$–iCre transgenic mouse line used in our study could have ChR2 channels delivered to peripheral

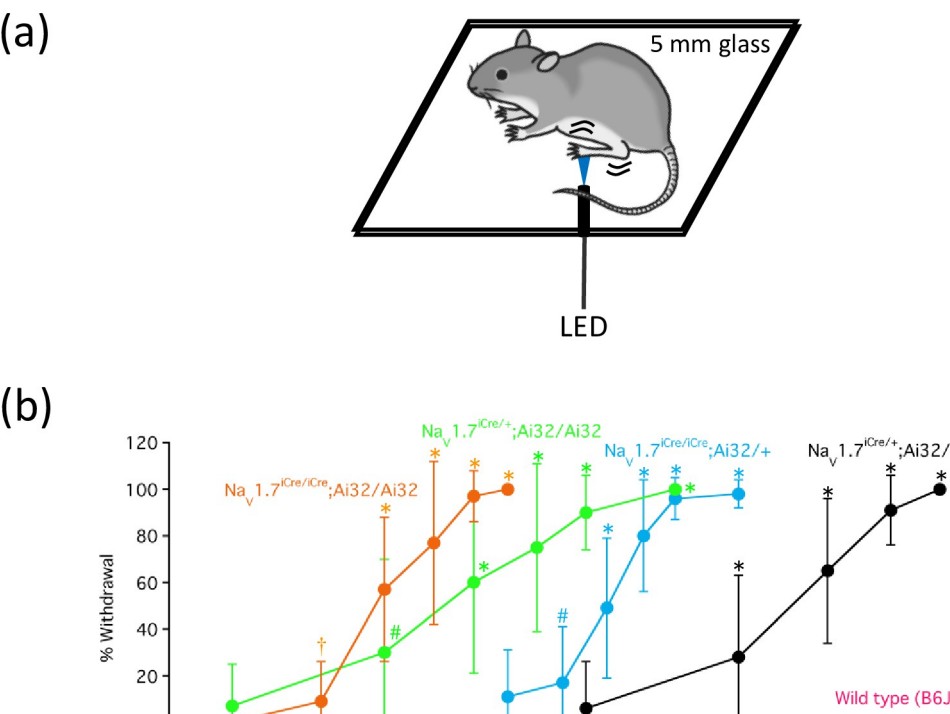

**Fig 4. Light irradiation hind paw withdrawal test.** (a) Experimental schematic. (b) The blue light irradiation hind paw withdrawal test was performed in wild-type (WT) and mice of the four genotypes. The data were analyzed using one-way ANOVA followed by Bonferroni post-hoc analysis. All results are calculated as mean ± SD of 10 or more animals. $^*P < 0.001$, compared with WT mice. $^†P = 0.03$ and $^#P = 0.02$, compared with WT mice.

nociceptors, allowing them to take advantage of the strong CAG promoter driving their conditional expression. This study also demonstrated that the genotype with a higher expression of ChR2 in the DRG showed a greater nocifensive response associated with blue light.

The first attempts to apply optogenetics to explore the underlying mechanisms of pain targeted Mas-related G-protein-coupled receptor D (Mrgprd) expressed in nonpeptidergic nociceptive C-fibers and transient receptor potential vanilloid-1 (TRPV1) expressed in peptidergic nociceptive C-fibers [15–18]. Transdermal optogenetic activation of TRPV1-expressing peripheral nociceptors induces nociceptive behaviors, including paw withdrawal and paw licking in mice, as well as conditioned place aversion, whereas the activation of Mrgprd-expressing neurons induces paw withdrawal and paw lifting, but not aversion [18].

The next target after Mrgprd and TRPV1 was $Na_V1.8$, a voltage-dependent $Na^+$ channel. Using the Cre-loxP strategy, Daou et al. crossed homozygous $Na_V1.8$–Cre mice with heterozygous Ai32 mice and generated a transgenic mouse line in which ChR2 was selectively targeted in $Na_V1.8$-expressing sensory neurons. In this line, ChR2 was expressed in the cell bodies as well as the fibers that reach the skin and the superficial layers of the spinal cord. Paw withdrawal was induced by blue light illumination of the hind paw skin of free-moving mice, thus demonstrating that they were the first transgenic mice to sense light as pain [11]. Furthermore, extended exposure of the hind paw to blue light (30 min under anesthesia) induced long-term behavioral sensitization to mechanical and thermal stimuli, and a 10-min exposure induced c-Fos expression in dorsal horn neurons ipsilateral to the stimulated hind paw [11]. Thus, the

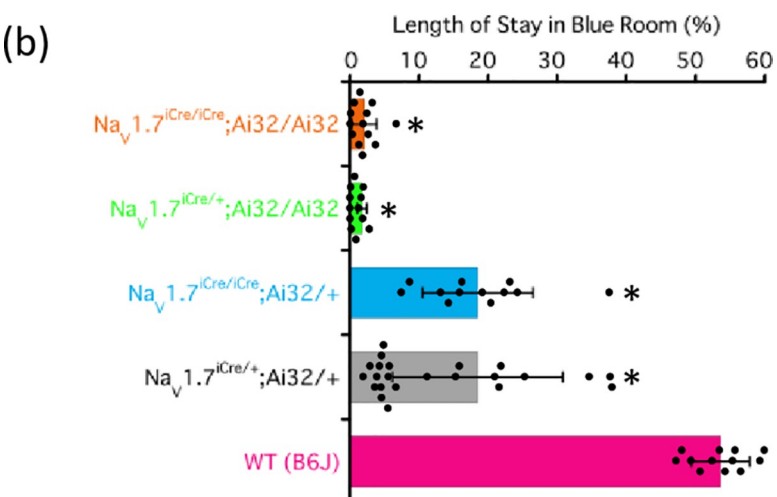

**Fig 5. OPA test.** (a) Experimental schematic. (b) The OPA test was performed with the wild-type (WT) and mice of the four genotypes. The data were analyzed using one–way ANOVA followed by Bonferroni post-hoc analysis. All results are calculated as mean ± SD of 10 or more animals. $^*P < 0.001$, compared with WT mice.

ability to evoke nocifensive behaviors with light alone provides a novel method of stimulation that is non-invasive, does not require mechanical interruption of the skin, and can be repeated without tissue injury. However, because $Na_V1.8$ currents are produced by $Na_V1.8$–Cre heterozygotes but not by $Na_V1.8$–Cre homozygotes, it is likely that the knock-in of the *Cre* gene affects the $Na_V1.8$ gene [19]. Thus, $Na_V1.8$–Cre homozygotes may have different pain responses to mechanical or thermal stimuli. The $Na_V1.7$–iCre mice in our study showed no difference in nocifensive response compared with that in WT mice, regardless of them being heterozygotes or homozygotes (S1 Fig). This is an advantage when $Na_V1.7$–iCre homozygotes are required to be included in experiments, as in this study.

As well as $Na_V1.8$, $Na_V1.7$ is also expressed in the peripheral nerve and play important roles in the development of inflammatory and neuropathic pain [9, 10]. Global $Na_V1.7$-null mutant mice die shortly after birth, whereas the $Na_V1.8$-null mutation is not lethal [20–22]. In $Na_V1.8$–Cre mice crossed with floxed $Na_V1.7$ mice, which are knockout mice specifically

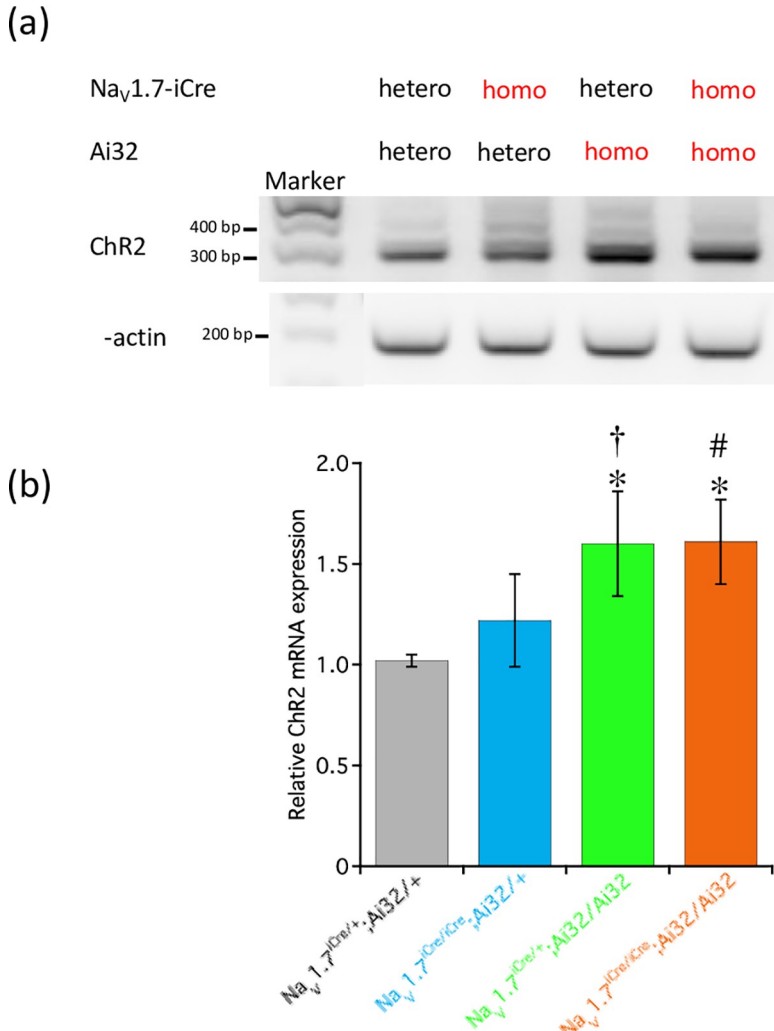

**Fig 6. RT-PCR for ChR2 expression in DRG.** ChR2 expression in DRG neurons as measured by RT-PCR. β–Actin was used as a positive control to confirm successful protein extraction and equal loading of samples. All data are calculated as mean ± SD of 5 animals. $^*P < 0.001$, compared with $Na_V1.7^{iCre/+}$;Ai32/+ mice. $^†P = 0.007$ and $^#P = 0.006$, compared with $Na_V1.7^{iCre/iCre}$;Ai32/+ mice.

lacking the nociceptor $Na_V1.7$, acute inflammatory pain responses evoked by various stimuli (e.g., carrageenan, Freund's adjuvant, or formalin) are reduced or abolished [22]. $Na_V1.7$ is the first voltage-dependent $Na^+$ channel to be identified with an altered functional mutation in humans [10]. Gain-of-function mutations in *Scn9a*, which encodes $Na_V1.7$, lead to severe neuropathic pain, whereas loss-of-function mutations in this gene lead to an indifference to pain. In the epidermis and spinal cord dorsal horn, $Na_V1.7$ is thought to enhance subthreshold stimuli, making it easier for neurons to reach the threshold for firing [23]. Thus, $Na_V1.7$ acts as an amplifier of the receptor potential in nociceptive neurons and plays a critical role in inherited erythromelalgia and paroxysmal extreme pain disorder, as it causes gain-of-function mutations that enable the channel to open with small depolarizations. Conversely, congenital insensitivity to pain is a loss-of-function mutation in $Na_V1.7$. $Na_V1.7$-deficient patients have no cognitive or cardiac function impairment; hence, analgesic therapies targeting $Na_V1.7$ have been investigated using these *Scn9a* mutations to generate global *Scn9a* knockout mice or rats

[24]. Furthermore, animal studies have shown that $Na_V1.7$ expression and function are increased in models of diabetic neuropathy, chronic constrictive injury, and chemotherapy-induced peripheral neuropathy [25]. These findings suggest that the pathophysiological mechanisms underlying these pain conditions can be understood by examining the profile of nociceptive behaviors related to $Na_V1.7$ and further comparing it to those associated with $Na_V1.8$ and $Na_V1.9$, which have also been implicated in chronic pain [21, 26]. With the development of genetic recombination technology, it will be easier to create animal models targeting these channels and thus, gain a deeper understanding of the role of these channels in various pain conditions. The present study showed that genotypes with high ChR2 expression (i.e. $Na_V1.7^{iCre/+}$;Ai32/Ai32 and $Na_V1.7^{iCre/iCre}$;Ai32/Ai32 mice) have a stronger nociceptive response to blue light because ChR2 expression is enhanced even in DRG neurons with low $Na_V1.7$ expression. However, since $Na_V1.7$ is widely expressed in various DRG neurons at varying expression levels, the enhanced expression of ChR2 did not seem to reflect such localization of $Na_V1.7$. Therefore, it is suggested that when comparing $Na_V1.7$ to $Na_V1.8$ or $Na_V1.9$, Cre and Ai32 should be compared in hetero phenotypes (i.e., $Na_V1.7^{iCre/+}$;Ai32/+ vs. $Na_V1.8^{iCre/+}$;Ai32/+ vs. $Na_V1.9^{iCre/+}$;Ai32/+).

In conclusion, we generated bicistronic $Na_V1.7$–iCre knock-in mice and developed a non-invasive pain model in which peripheral nociceptors were optically activated in free-moving transgenic $Na_V1.7$–ChR2 mice. An optogenetic approach to investigate the individual role of $Na^+$ channel subtypes in pain transduction might provide a better understanding of neuropathic pain development and help in designing effective therapeutics in the future.

## Methods

### Animals

WT C57BL/6J mice, commonly known as B6J mice, and Ai32 mice (C57BL/6 background) were purchased from the Jackson Laboratory (Bar Harbor, ME, USA). All the mice were individually housed in a temperature and humidity-controlled environment with a 12-h light-dark cycle and were permitted free access to food and water. This study was conducted in strict accordance with the guidelines for the Proper Conduct of Animal Experiments (Science Council of Japan) and approved by the Experimental Animal Care and Use Committee of the University of Miyazaki (Permit Number: 2018–536). Male mice over 2 months of age were used in the experiments. All efforts were made to minimize the number of animals used and their suffering. Mice in each group were randomly selected, and the experimenter was blinded to the mouse genotype.

### von Frey test to determine mechanical sensitivity

Mechanical sensitivity was examined by determining the paw withdrawal threshold using an electronic von Frey esthesiometer (IITC Life Science Inc., Woodland Hills, CA, USA) fitted with a polypropylene tip. Each adult mouse was placed in a 10 cm × 10 cm suspended chamber with a metallic mesh floor. After acclimation of mice for 30 min, the polypropylene tip was applied perpendicularly to the plantar surface of the right and left hind paws with sufficient force for 3–4 s. Brisk withdrawal or paw flinching was considered a positive response. The pain threshold was calculated as the mean of three measurements.

### Plantar test to determine thermal sensitivity

Thermal sensitivity was examined by measuring paw withdrawal latency in response to noxious thermal stimuli using plantar test (Hargreaves method) units (Ugo Basile SRL, Gemonio VA, Italy). Each mouse was placed on clear glass in an enclosure, and paw withdrawal latency

was measured. After a 30-min acclimation period, the heat-emitting projector lamp of the thermal test apparatus was activated, and the beam was directed to the plantar surface of the hind paw. Using a built-in digital timer, the paw withdrawal latency was recorded. The average withdrawal time over five consecutive trials was calculated. A cutoff value of 30 s was used to avoid possible tissue damage.

## Light irradiation test to determine paw withdrawal latency

Mice were habituated for 1 h in transparent cubicles (10 cm × 6.5 cm × 6.5 cm) set atop a 5-mm-thick glass floor and separated from each other by opaque dividers. Acute nocifensive behaviors were elicited light from using a pulsing LED light (465 nm blue light and 595 nm yellow light at 10 Hz; Doric Lenses Inc., Quebec, Canada) set at different intensities and aimed at the plantar surface of the hind paw. The light intensity was determined using a light power meter (LPM-100$^{TM}$; Bioresearch Center Inc., Aichi, Japan). Since the power meter measures light intensity in units of milliwatts (mW), the light density in units of mW/mm$^2$ was calculated by dividing the light intensity by the illuminated area in square millimeters (48 mm$^2$). The mice underwent a total of five trials of 1 s each, with 5-s intervals between the trials. The percentage of trials during which hind paw withdrawal or paw licking occurred was recorded.

## OPA test

This test was performed using an OPA system (Bioresearch Center Inc., Nagoya, Japan) [27]. We used a two-chamber system with an entrance connecting each chamber of 20 cm × 24 cm size. The floor of each chamber, one green (530 nm) and the other blue (470 nm), was illuminated with a 20 × 24 array of light-emitting diodes (LEDs). The chambers were uniformly illuminated (10 mW) to prevent the test being affected by the preference mice have for dark environments. After the mice were habituated to the chambers for 10 min with the LEDs turned off, each mouse was allowed to move freely between the two chambers for 10 min with the LEDs turned on. The location of each mouse was recorded using a video camera and analyzed using the BIOBSERVE Viewer 2 software. Next, the lights were switched on, and the position was recorded for 10 min. The percentage of time spent by each mouse in the blue- and green-floored chamber during the 10-min observational period was determined.

## RT-PCR

Mice were euthanized using sevoflurane exposure. Next, DRG of each mice genotype were obtained and immediately dissected for further analysis. Briefly, the collected DRG were homogenized, and total cellular RNA was isolated by acid guanidinium thiocyanate-phenol-chloroform extraction using TRIzol™ reagent (Total RNA Isolation Reagent; Invitrogen, Carlsbad, CA, USA). The quality and quantity of the extracted total cellular RNA were assessed by determining the ratio of the optical densities at 260 and 280 nm. The RT reaction was performed using a first-strand cDNA synthesis kit (SuperScript™ II Reverse Transcriptase, Invitrogen) according to the manufacturer's instructions. We performed PCR amplification using the EmeraldAmp$^{®}$ MAX PCR Master Mix (TAKARA Bio Inc., Shiga, Japan), 1 μL of cDNA template, and 0.4 μM of both forward and reverse primers in a 20-μL reaction mixture. The target cDNA was amplified according to the PCR protocol consisting of a denaturation step (10 min at 95˚), followed by 27 cycles (10 s at 98˚C, 30 s at 55˚C, and 60 s at 72˚C) for β-actin or 34 cycles (10 s at 98˚C, 30 s at 55˚C, and 60 s at 72˚C) for ChR2, and a final extension step (90 s at 72˚C). PCR was performed in a thermal cycler (Veriti™ Thermal Cycler; Thermo Fisher Scientific, Waltham, MA, USA). The PCR products were run on a 2% agarose gel. Subsequently, the bands were visualized using a luminoimage analyzer LAS-4000 (Fujifilm, Tokyo, Japan). We

used the following specific primers obtained from Macrogen Global Headquarters (Seoul, Korea); ChR2-Forward: 5′-caatgttactgtgccggatg-3′, ChR2-Reverse: 5′-attt-caatggcgcacacata-3′, β-actin-Forward: 5′-cgtaaagacctctatgccaaca-3′, β-actin-Reverse: 5′-cggactcatcgtactcctgct-3′ [28].

## Immunohistochemistry

The mice were anesthetized with sevoflurane. Next, they were intracardially perfused with 50 mL of perfusion buffer, followed by 100 mL of 4% paraformaldehyde (PFA) in phosphate-buffered saline (PBS) (pH 7.4) at room temperature for 30 min. DRG were dissected and post-fixed in 4% PFA for 2 h at 4˚C, cryoprotected in 30% sucrose in PBS, and incubated overnight at 4˚C. Sections of 14-μm thickness were cut from the freeze-fixed DRG with the temperature maintained at –20˚C using a cryostat (Leica Biosystems, Nussloch, Germany). For $Na_V1.7$ staining, sections of DRG were incubated in 0.1% Triton X-100 and 5% goat serum in PBS at room temperature for 4 h, followed by incubation with anti-$Na_V1.7$ polyclonal antibody (rabbit anti-rat, 1:250; catalog #ASC-008, Alomone Labs, Jerusalem, Israel) at 4˚C with overnight agitation. Subsequently, the sections were washed thrice with PBS, following by incubation with goat anti-mouse IgG (H+L) (Alexa Fluor™ 594, 1:500; catalog #ab150120, Abcam, Cambridge, MA, UK) for 1 h at room temperature. S Samples of spinal cord and hindpaw glabrous skin were placed directly on slides and sliced into 30-μm thick sections. These sections were washed, air-dried, and mounted with a coverslip using an antifade mounting medium. The fluorescence of the transgenic ChR2–EYFP on DRG, Spinal cord, and glabrous skin was sufficient to be visualized without immunostaining. The prepared slides were stored at 4˚C until further examination. The morphology of the different tissues was analyzed using a BZ-9000 fluorescence microscope (Keyence, Osaka, Japan). Images were processed using ImageJ software (NIH, Bethesda, MD, USA) to optimize brightness and contrast [29].

## Statistical analysis

Each behavioral experiment was evaluated $n \geq 10$ animals. For the behavioral experiments and RT-PCR, data were analyzed using one-way analysis of variance (ANOVA) followed by Bonferroni post-hoc analysis. The results are presented as mean ± SD. Statistical significance was set at $P < 0.05$. The statistical software JMP Pro 15 (SAS Institute, Inc., Cary, NC, USA) for Macintosh was used.

## Supporting information

**S1 Fig. von Frey test and plantar test of $Na_V1.7^{iCre/iCre}$ and $Na_V1.7^{iCre/+}$.** The von Frey test (a) and plantar test (b) were performed with wild-type (WT), $Na_V1.7^{iCre/iCre}$, and $Na_V1.7^{iCre/+}$ mice. The data were analyzed using one-way ANOVA. All results are calculated as mean ± SD of 10 or more animals. Individual results for each strain are (a) WT (B6J): 4.4 ± 0.7 g, $Na_V1.7^{iCre/iCre}$: 4.9 ± 0.8 g, and $Na_V1.7^{iCre/+}$: 4.9 ± 0.6 g, (b) WT (B6J): 5.7 ± 0.8 s, $Na_V1.7^{iCre/iCre}$: 5.5 ± 1.1 s, and $Na_V1.7^{iCre/+}$: 6.0 ± 1.8 s.
(PPTX)

**S2 Fig. Distribution of ChR2-EYFP channels in DRG.** The green fluorescent signal represents the direct fluorescence of ChR2-EYFP. (a) 20× and (b) 40×.
(PPTX)

**S1 Raw images. The original unadjusted and uncropped images of PCR presented in Fig 6A.**
(PDF)

## Acknowledgments

This study was conducted in the Department of Anesthesiology, Faculty of Medicine, University of Miyazaki. The authors would like to thank Noriko Hidaka, Kaori Kaji, Ayako Miura, and Hikaru Nakagawa for their technical and secretarial assistance in this study. The authors would like to thank Editage (www.editage.jp) for English language editing.

## Author Contributions

**Conceptualization:** Toyoaki Maruta.

**Data curation:** Toyoaki Maruta.

**Formal analysis:** Toyoaki Maruta, Yohko Kage, Tetsuro Shirasaka, Toshihiko Yanagita, Ryu Takeya, Isao Tsuneyoshi.

**Funding acquisition:** Toyoaki Maruta.

**Investigation:** Toyoaki Maruta, Kotaro Hidaka, Satoshi Kouroki, Tomohiro Koshida, Mio Kurogi.

**Methodology:** Toyoaki Maruta, Seiya Mizuno, Satoru Takahashi.

**Project administration:** Toyoaki Maruta.

**Resources:** Seiya Mizuno, Satoru Takahashi.

**Writing – original draft:** Toyoaki Maruta, Seiya Mizuno.

**Writing – review & editing:** Ryu Takeya, Isao Tsuneyoshi.

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
