## [Decision Letter · Decision Letter 0]

18 Jul 2022

PONE-D-22-16820Selective optogenetic activation of NaV1.7-expressing afferents in NaV1.7-ChR2 mice induces nocifensive behavior without affecting responses to mechanical and thermal stimuliPLOS ONE

Dear Dr. MARUTA,

Thank you for submitting your manuscript to PLOS ONE. After careful consideration, we feel that it has merit but does not fully meet PLOS ONE’s publication criteria as it currently stands. Therefore, we invite you to submit a revised version of the manuscript that addresses the points raised during the review process.

The reviewers are convinced of the usefulness of the novel tool you developed, however they asked for better characterization of both the mouse line (integration specificity, locus characterization, expression faithfulness, etc. ) and depth of behavior characterization. Please address all concerns of reviewers in your revised manuscript.

We look forward to receiving your revised manuscript.

Kind regards,

Tudor C. Badea, M.D., M.A., Ph.D.

Academic Editor

PLOS ONE

Journal Requirements:

Reviewers' comments:

Reviewer's Responses to Questions

**Comments to the Author**

1. Is the manuscript technically sound, and do the data support the conclusions?

Reviewer #1: Partly

Reviewer #2: Partly

2. Has the statistical analysis been performed appropriately and rigorously? 

Reviewer #1: I Don't Know

Reviewer #2: Yes

3. Have the authors made all data underlying the findings in their manuscript fully available?

Reviewer #1: Yes

Reviewer #2: Yes

4. Is the manuscript presented in an intelligible fashion and written in standard English?

Reviewer #1: No

Reviewer #2: Yes

5. Review Comments to the Author

Reviewer #1: In this study, the authors generated a new Nav-1.7-iCre knocking mice using CRISPR and crossed it to Ai32 reporter mice to have ChR2 expressed in Cre+ mouse DRG neurons. The authors then conducted VFH, thermal reflex, and peripheral light induced spontaneous behaviors and chamber preference assays. Overall, it's a new mouse line that is worth to be published, so the field is aware of the line and experimental results. However, there are major issues that the authors will need to fix during revision.

1) the manuscript title is misleading (or wrong). The mechanical and thermal tests (no deficits) have nothing to do with ontogenetic activation of Nav-1.7-Cre+ neurons. These are baseline behaviors. Thus, the title must be changed.

2) There is little data showing characterizing of the new Nav1.7-Cre line. The illustration showing the genomic structure, but no PCR or sequencing validating insert as anticipated. There is no co-staining of reporter (GFP) with endogenous Nav1.7 (or double in situ) or other common DRG neuron markers, CGRP, IB4, NF200. For the DRG section images, only GFP, and there is no quantification!!!

3) For peripheral light triggered paw withdrawal reflex, how do the authors know that it indicates pain response? Fos immunostaining with spinal cord sections after blue light stimuli would help to strengthen this point.

Other suggestion is to combine figures. Each current figure contains only one panel.

In addition to the experiments/data, the writing is also problematic. The description of generating Nav1.7 mouse line should be a main result section (the first part), which was put in the method instead. The results part overall is too short and over simplified. On the other hand, the discussion is lengthy and some parts (like Mrgd-ChR2 mice) are even irrelevant. The entire manuscript will need to be re-written.

Reviewer #2: The authors generated and characterized four genotypes of mice expressing ChR2 in Nav1.7-expressing primary afferent neurons. The four different genotypes show no difference compared to wild type animals in acute pain behaviour when exposed to von Frey filament stimulation and plantar test. However, the authors demonstrated nocifensive behaviour of these animals when stimulated with blue light, in a light intensity-dependent manner. The knock-in animals also displayed conditioned place aversion to blue light, as they spent less time in a room with a blue LED-illuminated floor, compared to a green LED-illuminated one. This work described a novel and valuable animal model to investigate further the role of the TTX-sensitive voltage-gated sodium channel Nav1.7 in a variety of pathological pain states. The experiments are clearly described and the conclusions are in line with the experimental results. This reviewer has only a few reservations concerning the general presentation of the experimental results and particularly the characterization of ChR-EYFP in DRG sections as presented in Fig. 6.

1. As a general observation, the Results section is extremely succinct, and the overall description of the data lacks quantitative detail. A more in-depth presentation of the actual results is warranted. For example, the authors limit their description of the nocifensive behaviour elicited by blue light to paw withdrawal (in Fig.3) and they do not discuss other manifestations which are generally associated with pain behaviour (paw licking, jumping, vocalization, etc…). Moreover, an important control is missing from the data shown in Fig.3: the experimenters should have used light of a different frequency (yellow light, at 590 nm, for example) at the highest light intensity used in the experiment, to rule out a ChR-independent effect of strong illumination on the animal behaviour. Finally, the light intensity used in the experiments illustrated in Fig.3 should be provided in mW/mm2, not in mW.

2. There is no attempt to quantify and characterise in more detail the expression of ChR-EYFP in DRG cell bodies in the sections illustrated in Fig.6. More information should be provided in terms of what percentage of neurons present YFP fluorescence, what is their average size, etc… It would also be of interest to dissect the expression of ChR-EYFP in peptidergic versus non-peptidergic neurons, using an anti-CGRP and/or an anti-P2X3 antibody, for instance.

3. It would be of interest to investigate and confirm the trafficking of ChR to peripheral and central endings of Nav1.7-expressing peripheral nociceptors, by monitoring YFP fluorescence in skin and spinal cord sections.

6. PLOS authors have the option to publish the peer review history of their article (what does this mean?). If published, this will include your full peer review and any attached files.

Reviewer #1: No

Reviewer #2: No

---

## [Author Response · Author response to Decision Letter 0]

31 Aug 2022

Dear Dr. Tudor C. Badea, 

Thank you for giving me the opportunity to submit a revised draft of my manuscript titled “Selective optogenetic activation of NaV1.7–expressing afferents in NaV1.7-ChR2 mice induces nocifensive behavior without affecting responses to mechanical and thermal stimuli” to PLOS ONE (Manuscript ID: PONE-D-22-16820). We appreciate the time and effort that you and the reviewers have dedicated to providing your valuable feedback on our manuscript. We are grateful to the reviewers for their insightful comments on my paper. 

We have extensively revised our paper to reflect most of the suggestions provided by the reviewers. We have highlighted the changes within the manuscript in red font. Here is a point-by-point response to the reviewers’ comments and concerns.

Reviewer #1

1) the manuscript title is misleading (or wrong). The mechanical and thermal tests (no deficits) have nothing to do with ontogenetic activation of NaV1.7-Cre+ neurons. These are baseline behaviors. Thus, the title must be changed.

Response: The transgenic mice used in our study have a knock-in of the iCre recombinase gene downstream of Scn9a in the NaV1.7 gene. Therefore, we were concerned about the normal expression and functioning of NaV1.7. As mentioned in the discussion section (Page 24, lines 263-270), NaV1.8 currents were not observed in previously generated NaV1.8-Cre homozygote mice. Therefore, the nociceptive response to mechanical and thermal stimuli was likely attenuated. To address this concern, we wanted to demonstrate that the nociceptive response to mechanical and thermal stimuli, which involves functioning of the NaV1.7, is at least normal. Therefore, we included the phrase “without affecting responses to mechanical and thermal stimuli” in the manuscript title.

2) There is little data showing characterizing of the new NaV1.7-Cre line. The illustration showing the genomic structure, but no PCR or sequencing validating insert as anticipated. There is no co-staining of reporter (GFP) with endogenous NaV1.7 (or double in situ) or other common DRG neuron markers, CGRP, IB4, NF200. For the DRG section images, only GFP, and there is no quantification!!!

Response: We omitted the quantification of ChR2 in the immunostaining of DRG because we used PCR to assess the expression of ChR2 in the DRG of each phenotype. Immunostaining images of “Distribution of ChR2-EYFP channels in DRG” for mice of the four genotypes has been provided as supplementary information (S2 Fig). In addition, we added a figure showing NaV1.7 and ChR2-EYFP in the DRG of NaV1.7iCre/+;Ai32/+ mouse (Fig 2b). We revised the text in the manuscript accordingly (Page 14: lines 147-151).

“Fig 2b shows a typical immunohistochemical image showing NaV1.7 and ChR2-EYFP in the DRG of NaV1.7iCre/+;Ai32/+ mouse. EYFP fluorescence was observed in NaV1.7-expressing DRG neurons. EYFP fluorescence was also observed in the dorsal horn of the spinal cord and was faded in the deep and superficial layers of the glabrous skin bordering the dermal– epidermal junction.”

Although staining of common DRG neuron markers was not performed in this study, we intend to perform such experimentation in the future.

3) For peripheral light triggered paw withdrawal reflex, how do the authors know that it indicates pain response? Fos immunostaining with spinal cord sections after blue light stimuli would help to strengthen this point.

Response: We assessed nocifensive behavior and did not focus on proving whether the mice strictly felt pain. We believe that the transgenic mice developed in this study is a mouse model for pain; however, we have differentiated between “nocifensive” and “painful” as much as possible. 

Examining the changes in Fos expression induced by short-term light irradiation would be difficult. For instance, Daou et al. demonstrated the ipsilateral c-Fos expression in neurons of laminae I–III of the dorsal horn after 10 min suprathreshold blue-light stimulation of the left hindpaw of NaV1.8 – ChR2+ mice (J. Neurosci. 2013; 33: 18631–18640). Currently, we are investigating the persistence of nociceptive behavior after prolonged light exposure and considering the examination of changes in pain-related molecule expressions in the spinal cord and DRG.

4) Other suggestion is to combine figures. Each current figure contains only one panel.

Response: The number of figures has been reduced to six in total, because the old Fig. 1 and 2 were combined into new Fig. 3 and the old Fig. 6 was changed to S2 Fig.

5) In addition to the experiments/data, the writing is also problematic. The description of generating NaV1.7 mouse line should be a main result section (the first part), which was put in the method instead. The results part overall is too short and over simplified. On the other hand, the discussion is lengthy and some parts (like Mrgd-ChR2 mice) are even irrelevant. The entire manuscript will need to be re-written.

Response: The sections "Development of original NaV1.7–iCre (NaV1.7iCre/+) knock-in mice" and "Generation of the four genotypes of NaV1.7-ChR2 mice" were moved to the Results section (Pages 9-12, lines 92-128) from the Methods section.

To address the issue of a lengthy discussion, the discussion regarding Mrgprd-ChR2 and TRPV1-ChR2 mice in the discussion section has been shortened (Pages 22-23, lines 243-250).

 Reviewer #2

1) As a general observation, the Results section is extremely succinct, and the overall description of the data lacks quantitative detail. A more in-depth presentation of the actual results is warranted. For example, the authors limit their description of the nocifensive results is warranted. For example, the authors limit their description of the nocifensive behaviour elicited by blue light to paw withdrawal (in Fig.3) and they do not discuss other manifestations which are generally associated with pain behaviour (paw licking, jumping, vocalization, etc...). Moreover, an important control is missing from the data shown in Fig.3: the experimenters should have used light of a different frequency (yellow light, at 590 nm, for example) at the highest light intensity used in the experiment, to rule out a ChR-independent effect of strong illumination on the animal behaviour. Finally, the light intensity used in the experiments illustrated in Fig.3 should be provided in mW/mm2, not in mW.

Response: We did not assess other nociceptive responses separately in detail in this study. We did not observe jumping and vocalization by mice, which are generally associated with nocifensive behavior, during experimentation. Paw licking was occasionally observed and was considered a positive nocifensive behavior, as was paw withdrawal. Therefore, the method text was revised as follows.

Page 32, Lines 354-355: “The percentage of trials during which hind paw withdrawal or paw licking occurred was recorded.”

Indeed, this comment by reviewer #2 made us realize the importance of eliminating the ChR2-independent effects of strong illumination on animal behavior. Unfortunately, phenotypes other than NaV1.7iCre/+;Ai32/+ mice are not maintained due to breeding space issues. Consequently, an extended amount of time would be required to obtain a suitable number of these mice phenotypes other than NaV1.7iCre/+;Ai32/+ for this control experiment. Hence, as a control experiment, we exposed the NaV1.7iCre/+;Ai32/+ mice to yellow light. This was reflected in the manuscript by adding the following sentences in the Results section.

Pages 17-18, Lines 191-195: “To rule out a ChR2-independent effect of strong illumination on the animal behavior, we performed a yellow LED light irradiation hind paw withdrawal test on NaV1.7iCre/+;Ai32/+ mice. Irradiation with yellow light irradiation (5 mW) did not induce hind paw withdrawal behavior.”.

The light power meter that we used (LPM-100™; Bioresearch Center Inc., Aichi, Japan) displays light intensity in units of “mW”; to convert it to “mW/mm2”, we divided the light intensity by the area of illumination (48 mm2). Accordingly, the added the following sections to the methods section. 

Page 31, Lines 351-353: “Since the power meter measures light intensity in milliwatts (mW), the light density in mW/mm2 was calculated by dividing the light intensity by the illuminated area in square millimeters (48 mm2).”

2) There is no attempt to quantify and characterise in more detail the expression of ChR-EYFP in DRG cell bodies in the sections illustrated in Fig.6. More information should be provided in terms of what percentage of neurons present YFP fluorescence, what is their average size, etc... It would also be of interest to dissect the expression of ChR-EYFP in peptidergic versus non-peptidergic neurons, using an anti-CGRP and/or an anti-P2X3 antibody, for instance.

Response: We omitted the quantification of ChR2 in the immunostaining of DRG because we used PCR to assess the expression of ChR2 in DRG of each phenotype. Immunostaining images of “Distribution of ChR2-EYFP channels in DRG” for mice of the four genotypes has been provided as supplementary information (S2 Fig). As reviewer #2 pointed out, it would be interesting to characterize ChR2 expression using anti-CGRP and anti-P2X3 antibodies. Currently, we are developing mice targeting NaV1.8 and NaV1.9 as well as NaV1.7-ChR2 mice and will report on this subject in our next study.

3) It would be of interest to investigate and confirm the trafficking of ChR to peripheral and central endings of NaV1.7-expressing peripheral nociceptors, by monitoring YFP fluorescence in skin and spinal cord sections.

Response: We monitored the EYFP fluorescence in the dorsal horn spinal cord and hindpaw glabrous skin (Fig. 2b). Page 14, lines 149-151: “EYFP fluorescence was also observed in the dorsal horn of the spinal cord and was faded in the deep and superficial layers of the glabrous skin bordering the dermal– epidermal junction.”

---

## [Decision Letter · Decision Letter 1]

22 Sep 2022

Selective optogenetic activation of NaV1.7-expressing afferents in NaV1.7-ChR2 mice induces nocifensive behavior without affecting responses to mechanical and thermal stimuli

PONE-D-22-16820R1

Dear Dr. MARUTA,

We’re pleased to inform you that your manuscript has been judged scientifically suitable for publication and will be formally accepted for publication once it meets all outstanding technical requirements.

Kind regards,

Tudor C. Badea, M.D., M.A., Ph.D.

Academic Editor

PLOS ONE

Additional Editor Comments (optional):

Reviewers' comments:

Reviewer's Responses to Questions

**Comments to the Author**

1. If the authors have adequately addressed your comments raised in a previous round of review and you feel that this manuscript is now acceptable for publication, you may indicate that here to bypass the “Comments to the Author” section, enter your conflict of interest statement in the “Confidential to Editor” section, and submit your "Accept" recommendation.

Reviewer #2: All comments have been addressed

2. Is the manuscript technically sound, and do the data support the conclusions?

Reviewer #2: Yes

3. Has the statistical analysis been performed appropriately and rigorously? 

Reviewer #2: Yes

4. Have the authors made all data underlying the findings in their manuscript fully available?

Reviewer #2: Yes

5. Is the manuscript presented in an intelligible fashion and written in standard English?

Reviewer #2: Yes

6. Review Comments to the Author

Reviewer #2: The authors have responded in a satisfying manner to this reviewer's queries. The paper can be published in the revised form.

7. PLOS authors have the option to publish the peer review history of their article (what does this mean?). If published, this will include your full peer review and any attached files.

Reviewer #2: No

---

## [Editor Report · Acceptance letter]

27 Sep 2022

PONE-D-22-16820R1 

Selective optogenetic activation of NaV1.7–expressing afferents in NaV1.7-ChR2 mice induces nocifensive behavior without affecting responses to mechanical and thermal stimuli 

Dear Dr. Maruta:

I'm pleased to inform you that your manuscript has been deemed suitable for publication in PLOS ONE. Congratulations! Your manuscript is now with our production department. 

Kind regards, 

on behalf of

Dr. Tudor C. Badea 

Academic Editor

PLOS ONE